# (Cr_1−x_Al_x_)N Coating Deposition by Short-Pulse High-Power Dual Magnetron Sputtering

**DOI:** 10.3390/ma15228237

**Published:** 2022-11-20

**Authors:** Alexander Grenadyorov, Vladimir Oskirko, Alexander Zakharov, Konstantin Oskomov, Andrey Solovyev

**Affiliations:** The Institute of High Current Electronics SB RAS, 2/3, Akademichesky Ave., 634055 Tomsk, Russia

**Keywords:** CrAlN coating, hardness, wear resistance, dual-HiPIMS with short pulses, 100Cr6 counter body, Al_2_O_3_ counter body

## Abstract

The paper deals with the (Cr_1−x_Al_x_)N coating containing 17 to 54 % Al which is deposited on AISI 430 stainless steel stationary substrates by short-pulse high-power dual magnetron sputtering of Al and Cr targets. The Al/Cr ratio in the coating depends on the substrate position relative to magnetrons. It is shown that the higher Al content in the (Cr_1−x_Al_x_)N coating improves its hardness from 17 to 28 GPa. Regardless of the Al content, the (Cr_1−x_Al_x_)N coating manifests a low wear rate, namely (4.1–7.8) × 10^−9^ and (3.9–5.3) × 10^−7^ mm^3^N^−1^m^−1^ in using metallic (100Cr6) and ceramic (Al_2_O_3_) counter bodies, respectively. In addition, this coating possesses the friction coefficient 0.4–0.7 and adhesive strength quality HF1 and HF2 indicating good interfacial adhesion according to the Daimler-Benz Rockwell-C adhesion test.

## 1. Introduction

Hard and wear-resistant coatings TiN, TiAlN, CrAlN are widely used to improve the service life of different systems and mechanisms such as cutting and molder tools, friction assemblies, etc. [1,2,3]. In comparison with TiN and TiAlN coatings, the CrAlN coating possesses higher resistance to oxidation at high temperatures (because both chromium and aluminum can form protective oxides, which suppress the oxygen diffusion in the bulk [4,5,6]), and improved wear and corrosion resistance [4].

The CrAlN coating is usually obtained by physical vapor deposition, the widely used arc ion plating (AIP) and cathodic arc evaporation (CAE), which provide the coating with high hardness (25–40 GPa) and adhesive strength due to the high ionization degree of the evaporated material [7,8,9]. A major shortcoming of the AIP and CAE methods is the droplet fraction in the coating, which makes the its surface rougher and degrades the performance properties [10,11,12]. Magnetron sputtering is used as an alternative to the above techniques, which produces the coating without the droplet fraction and with low surface roughness [11]. The CrAlN coating is therefore obtained by ion-beam-enhanced magnetron sputtering [13], closed-field unbalanced magnetron sputtering [14,15,16,17], single/dual RF magnetron sputtering [3,18,19,20], direct-current (DC) magnetron sputtering [21,22], high-power impulse magnetron sputtering (HiPIMS) [23], continuous high-power magnetron sputtering [24], and combined high-power impulse magnetron (C-HPMS) and DC magnetron sputtering [25,26]. The HiPIMS technology has distinct advantages with respect to the coating adhesion, hardness and density [27,28].

Much research has been done thus far to study the effect from the substrate bias voltage [13], nitrogen content in the plasma and working pressure [19], high-temperature annealing [14,21,29], and Al/Cr ratio [15] on the structure, mechanical and tribological properties of the CrAlN coating. In the vast majority of cases, CrAl cathodes with either similar Cr and Al content [21,24,29,30] or dominant Al content [17,20,25] are used for the CrAlN coating deposition.

Weirather et al. [31] used segmented targets to modify the Cr and Al content in the CrAlN coating, which significantly reduced the coating development efforts. Metallic Al and Cr segments as well as Cr/Al (50/50) segments were combined in pairs to synthesize hard CrAlN coatings resulting in a wide range of the Al content. Almost the same method of changing the chemical composition of CrAlN coatings was used in [32], where the Al content ranged from 5 to 77 at.%. In these papers, the elemental composition was changed by the replacement of targets. However, according to PalDey and Deevi [33], it is important to control the elemental composition of coatings directly during the deposition process in order to obtain a gradient composition or multilayer coatings.

In depositing the (Cr_1−x_Al_x_)N coating, we offer to employ the dual magnetron sputtering system with a closed magnetic field, elemental targets of pure Cr and pure Al, and a power supply characterized by high-power short duration pulses. The proposed method has the following advantages. Firstly, the dual magnetron sputtering system allows the elimination of the problem of the anode disappearance during the deposition of low-conductivity coatings, thereby providing a long, stable deposition process. Secondly, two unbalanced magnetrons neighboring each other with opposite polarities can form a closed configuration of magnetic field, extending the plasma to the substrate and increasing the current density on it, providing a positive effect on the coating properties [34]. Thirdly, elemental targets of pure Cr and pure Al and discharge power control could be used to modify the coating composition over a wide range and generate coatings with a thickness gradient composition. Fourthly, high-power short duration pulses increase the ionization rate of the sputtered material, and the substrate bias voltage controls the energy of particles forming the coating. In our previous research [35] into dual high-power impulse magnetron sputtering, we proposed a power supply system which provided an independent control for bipolar pulse parameters. In addition, we demonstrated that the ion flux and ion-to-atom ratio on the substrate were higher in the HiPIMS mode with short pulses (10–20 μs) [36]. Tiron et al. [37] demonstrated that the ionized flux fraction during HiPIMS gradually increased as the pulse duration decreased to 4 μs.

Using the proposed technique, a (Cr_1−x_Al_x_)N coating with varying Al content is obtained in this work. The aspects studied are the impact of the Al/Cr ratio on mechanical and tribological properties of the coating, and wear mechanisms based on tribology tests with metallic (100Cr6) and ceramic (Al_2_O_3_) counter bodies.

## 2. Materials and Methods

### 2.1. Preparation of (Cr_1−x_Al_x_)N Coatings

The coating was deposited onto 20 × 20 mm^2^ substrates 0.5 mm thick made of polished AISI 430 stainless steel. The surface roughness was 0.08 ± 0.01 μm. Prior to the coating deposition, the substrates were cleaned ultrasonically in isopropyl alcohol and acetone. In each liquid, the substrates were treated for 5 min.

A dual magnetron sputtering system with Al and Cr cathodes, both of 99.95% purity and a diameter 76 mm, was used for the coating deposition. The flow-chart of the proposed dual high-power impulse magnetron sputtering (dual-HiPIMS) vacuum system is illustrated in Figure 1 together with the arrangement of S1–S4 substrates on the holder relative to magnetrons. The working chamber was vacuumized by a turbo-molecular pump to a residual pressure of 8 × 10^−3^ Pa.

A ceramic infra-red heater provided the substrate heating up to 450 ± 10 °C, which was maintained during the coating deposition. The (Cr_1−x_Al_x_)N coating deposition was conducted in a mixture of argon and nitrogen at a total pressure of 0.1 Pa with the flow rates of 83 and 15 sccm, respectively. The magnetrons were powered by a single bipolar power supply, APEL-M-10HPP-1500 (“Applied Electronics”, Tomsk, Russia), that had the ability to independently adjust the parameters of positive and negative pulses [36]. This allowed us to independently control the power of each magnetron. The power supply was operated in power stabilization mode. The average discharge power *P*_d.avg_ was estimated as
(1)$$P_{d.\mathrm{avg}}=\frac{1}{T}\int_{0}^{T}U_{d}I_{d}dt$$

The operating discharge parameters included 5 kHz frequency, 20 µs pulse duration, and 0.5 kW and 1 kW power, respectively, for sputtering Cr and Al targets. The maximum voltages for Cr and Al magnetrons were 694 V and 677 V, and the maximum currents were 30 and 60 A, respectively. Values of nitrogen flow rate and magnetron powers had been determined in preliminary experiments. The power of the Cr magnetron was chosen to be half as high because the rate of CrN deposition is about twice as high as the rate of AlN deposition. To determine the optimal nitrogen flow rate, (Cr_1−x_Al_x_)N coatings were obtained at different nitrogen flow rates (11, 13, 15 and 18 sccm) at the same argon flow rate and magnetron powers. The coatings were obtained on samples placed in the center of the substrate holder, i.e., at the same distance from both magnetrons. Then, the hardness of these coatings was measured. The maximum coating hardness (25 GPa) was obtained with a nitrogen flow rate of 15 sccm. Therefore, further experiments were conducted at this nitrogen flow rate.

To improve the coating adhesive strength and provide its surface activation, the bipolar bias voltage with negative pulses of 800 V, 100 kHz and 70% duty cycle was supplied to the substrate for the first 5 min. The positive pulse amplitude ranged between 15 and 20% of the negative pulse amplitude. The deposition was conducted at 100 V negative pulse amplitude for 120 min. The coating thickness and the deposition rate were measured on an interference microscope, MII-4 (LOMO, St. Petersburg, Russia).

### 2.2. Surface Characterization

Determination of the elemental composition of the (Cr_1−x_Al_x_)N coating and wear scar observation was carried out with a Quanta 200 3D (FEI Company, Hillsboro, OR, USA) scanning electron microscope (SEM) coupled with an energy dispersive X-ray analyzer.

X-ray diffraction (XRD) patterns of the coatings were obtained using a Shimadzu XRD-6000 Diffractometer (Kyoto, Japan). Measurements were conducted using Cu *K*_α_ radiation in grazing-incidence X-ray diffraction geometry. Operating parameters for the XRD included 3.0° incidence angle, 20–80° scan range and 0.02° step angle. The analysis of the phase composition was performed using the PDF4+ database and the PowderCell 2.4 Rietveld program.

A NanoTest 600 hardness tester (Micro Materials Ltd., GB, Wrexham, UK) was used to measure the coating hardness by the Oliver and Pharr method [38]. The process parameters for the NanoTest 600 were 10 mN load, 20 s loading–unloading time, and 10 s maximum exposure time. Each substrate was indented 10 times, and the obtained results were averaged.

The tribological properties of the coatings were examined using the ball-on-disc method. Testing conditions included a 100 mm/s sliding speed, 2 N normal load applied to the steel (100Cr6) and ceramic (Al_2_O_3_) balls of the diameter 6 mm. Their physical and mechanical properties are given in Table 1. The wear track diameters for the steel and ceramic balls were 8 and 6 mm, respectively. At the end of the test, we measured the wear track profile with the interference MNP-2 (Russia) microscope-profilometer and the 130 (Zavod PROTON, Moscow, Russia) contact profilometer. The cross-sectional area of the wear track was calculated using the Origin 9 software. A Polar-1 metallographic microscope (Mikromed, Moscow, Russia) was used to examine the wear track in the reflected polarized light.

The wear volume is obtained from [39]:(2)$$V=\frac{\pi \cdot {\left(R-\sqrt{R^{2}-{\left(d/2\right)}^{2}}\right)}^{2}\cdot \left(2R+\sqrt{R^{2}-{\left(d/2\right)}^{2}}\right)}{3}$$
where *R* is the ball radius, and *d* is the wear track diameter. The wear rate results from the ratio between the wear volume for the disk/ball and the product of normal force by total sliding distance, viz. mm^3^·N^−1^·m^−1^.

A Rockwell-C scale hardness test conducted in accordance with the VDI 3198 standard [41] evaluated the adhesive strength of the coating. A TK-2 spheroconical diamond indenter (IvMashprom, Yekaterinburg, Russia) with a 120° included-cone angle was used for nanoindentation examinations. The full indentation load was 60 kg (588 N). The testing cycle continued for 2 s. After testing, the coating adhesion to the substrate ranged from perfect (HF1) to poor adhesion (HF6), depending on the number of cracks and delamination.

## 3. Results and Discussion

### 3.1. Chemical Composition

The rate of the (Cr_1−x_Al_x_)N coating deposition onto the stationary substrate depends on its position relative to magnetrons. Figure 2 presents the deposition rate distribution lengthwise over a glass substrate 11 cm long. Since the magnetron power with the Al target is 2 times higher than the magnetron power with the Cr target, the deposition rate near the former is maximum (48 ± 3 nm/min) and gradually lowers to 30 ± 2 nm/min as the Cr target approaches. These rates differ from each other by less than 2 times as aluminum has a much lower sputtering yield than chromium. The formation of the AlN insulating layer on the Al target surface during the reactive sputtering process reduces the sputtering yield of aluminum.

The results of the chemical analysis of substrates S1, S2, S3 and S4 arranged as presented in Figure 1b are summarized in a table in Figure 2. The Al content in the (Cr_1−x_Al_x_)N coating reduces from 54 at.% in substrate S1 to 17 at.% in substrate S4. The Al/Cr ratio grows by 6 times (from 0.2 to 1.2), depending on the substrate position. A small amount of oxygen (1–3 at.%) was also detected in the coatings.

### 3.2. Phase Composition of the (Cr_1−x_Al_x_)N Coating

The XRD patterns in Figure 3 obtained for the (Cr_1−x_Al_x_)N coating show that only peaks with (111), (200), (220) and (311) orientations matching the NaCl (B1) structure are present in substrate S4 (with the lowest Al content). This indicates that the crystal structure of the (Cr_1−x_Al_x_)N coating is mostly cubic, similar to the CrN coating structure. 

Substrate S4 is characterized by the highly (111)-oriented structure. For the substrate S4, a slight shift of the peak (111) probably relates to the residual stress. A structural evolution is clearly observed with increasing Al content in this coating. The structure orientation fully changes to (220) orientation with increasing Al content from 17 to 31% and remains so at higher Al concentrations. The diffraction peak positions shift toward the larger-angle region, indicating a dissolution of added aluminum to the CrN lattice, which substitutes Cr atoms. In this case, the lattice parameter reduces from 0.4155 nm for substrate S4 to 0.4111 nm for substrate S1. 

Banakh et al. [42] also reported on the preferred orientation (111) for the sputter-deposited (Cr_1−x_Al_x_)N coating with the low (x < 0.27) Al content. The preferred orientation (220) was observed for the coating with the higher Al content.

### 3.3. Mechanical Properties of the (Cr_1−x_Al_x_)N Coating

The block diagram in Figure 4a describes the hardness and elastic modulus of the (Cr_1−x_Al_x_)N coating depending on the substrate position and loading–unloading curves for this coating. Its thickness is over 2 µm and the maximum indentation depth is 110 nm (see Figure 4b), i.e., less than 10% of the coating thickness. In this case, the substrate does not significantly affect the hardness. With increasing Al content, the hardness and elastic modulus tend to grow. Thus, when the Al content increases from 17 to 54 at.%, the hardness grows from 17 to 28 GPa, and the elastic modulus increases from 173 to 318 GPa. The behavior of these two parameters can also be found in the literature [43,44]. The hardness is observed to grow with Al content increasing up to ~60 at.%. This can be explained by the strain-induced dissolution of Al atoms in the cubic lattice of the CrN coating, which hinders the dislocation movement [45]. When the Al content exceeds 63 at.%, the coating hardness lowers due to the formation of the hexagonal aluminum nitride (h-AlN).

As can be seen from Figure 4b, substrate S1 does not tend to creep during the pause running for 10 seconds as the displacement of the diamond indenter does not grow under the maximum constant load. At the same time, the coating low in aluminum manifests a creep as in the case with stainless steel substrates without the nitride coating [46].

Mechanical parameters of the (Cr_1−x_Al_x_)N coating calculated from nanoindentation are presented in Table 2. The resistance to plastic deformation *H*^3^/*E*^2^ and elastic recovery *W*_e_ of substrates S1, S2 and S3 reduce with decreasing Al content. No direct dependence is observed between the plasticity index *H*/*E* and the Al content. Although substrate S4 shows the highest plasticity index of ~0.1, its hardness is lower than that of other substrates. As shown in [47,48,49,50], the higher values of *H*/*E* and *H*^3^/*E*^2^ correlate with the high wear resistance of materials. It is thus possible to assume that the (Cr_1−x_Al_x_)N coating deposited on substrates S1 and S2 must possess the highest wear resistance.

### 3.4. Wear Resistance of (Cr_1−x_Al_x_)N/100Cr6 Sliding Pair

Tribology tests of the (Cr_1−x_Al_x_)N coating utilized steel (100Cr6) and ceramic (Al_2_O_3_) counter bodies of the diameter 6 mm. The wear rate was measured both for the coating and counter bodies. The block diagram in Figure 5 plots the results of the wear rate vs. the 100Cr6 counter body. One can see that the wear rate grows by 2 times with decreasing Al content, which correlates with the coating hardness reduction. The same dependence is observed in [51] for the (Cr_1−x_Al_x_)N/100Cr6 sliding pair. Nevertheless, all the substrates are characterized by the low wear rate of 10^−9^ mm^3^N^−1^m^−1^. As we assumed, substrates S1 and S2 have the lowest wear rate due to the high plasticity index and resistance to plastic deformation.

As noted in [2,52], one more parameter can fairly correctly predict the wear resistance of the substrate–coating system, namely the ratio *E*_s_/*E*_c_ between the elastic modulus of these system components. The highest wear resistance is observed when the ratio *E*_s_/*E*_c_ tends to unity. Huang et al. [2] show that the TiN coating demonstrates the highest wear resistance on Cu and M2 high-speed steel substrates. The TiN coating has a lower *E*_s_/*E*_c_ ratio (2.14 and 1.19 for Cu and M2, respectively) than the TiAlN coating (3.25 and 1.8 for Cu and M2, respectively). On the hard-alloy substrate WC–8 wt.% Co, the highest wear resistance belongs to the TiAlN coating with *E*_s_/*E*_c_ = 0.97, whereas for the TiN coating this ratio is 0.64. Łępicka et al. [46] confirm the correctness of using the *E*_s_/*E*_c_ ratio to predict the coating wear resistance. They report that the TiN coating on the 316LVM stainless steel (*E*_s_/*E*_c_ = 1.8) manifests a higher wear resistance than this coating on the Ti6Al4V alloy (*E*_s_/*E*_c_ = 3.06).

The elastic modulus of AISI 430 steel is 182 GPa. The ratio *E*_s_/*E*_c_ therefore reduces from 1.75 (for substrate S1) to 0.95 (for substrate S4), depending on the (Cr_1−x_Al_x_)N coating composition. Substrate S4 must possess the highest wear resistance, but our experiment shows that it lowers with decreasing *E*_s_/*E*_c_ ratio. This is probably associated with the counter body material. According to Table 1, the 100Cr6 steel ball has low hardness and resistance to plastic deformation, which leads to its intensive wear at a rate of ~2 × 10^−5^ mm^3^N^−1^m^−1^, as illustrated in Figure 5. Thus, in the case of the hard (Cr_1−x_Al_x_)N/soft 100Cr6 sliding pair, the ratio *E*_s_/*E*_c_ does not enable correct prediction of the wear resistance of hard nitride coatings.

Figure 6 contains SEM images of wear scars on the 100Cr6 ball surface. For all (Cr_1−x_Al_x_)N coatings, significant wear of the 100Cr6 ball surface is observed. The wear scar diameter ranges within 1200 and 1250 µm and weakly depends on the coating composition. One can see unidirectional scratches relative to sliding of the coating material particles.

In Figure 7, one can see wear tracks on the (Cr_1−x_Al_x_)N coating surface.

These tracks are produced by the particles of the counter body material smeared on the coating surface. The surface wear occurs by the adhesion mechanism. Wear tracks left by the steel counter body during friction on substrates S1–S4 are characterized by regions high in iron. Shallow grooves left by the material particles of the worn counter body are observed on substrate S4. This material reacts with oxygen and generates harder oxide particles.

The presence of iron and oxygen in the wear tracks is confirmed by the EDX analysis presented in Figure 8. Maps of the element distribution show that oxygen locates together with iron. This fact confirms oxidation of the worn material during tribochemical wear. The same is observed in [46,52] for the TiN coating during tribology testing.

### 3.5. Wear Resistance of (Cr_1−x_Al_x_)N/Al_2_O_3_ Sliding Pair

The block diagram in Figure 9 plots the results of the wear rate vs. the Al_2_O_3_ counter body. In this case, the wear rate increases by two orders of magnitude and is 10^−7^ mm^3^N^−1^m^−1^. The wear resistance tends to grow from 5.3 × 10^−7^ to 3.9 × 10^−7^ mm^3^N^−1^m^−1^ with decreasing Al content in the coating. This is because the harder coating on substrates S1 and S2 provides stronger wear of the counter body, and hard abrasive particles removed from the coating surface intensify its wear. The wear rate of the ceramic counter body on substrate S1 is 9.4 × 10^−8^ mm^3^N^−1^m^−1^, whereas on substrate S4 it reduces down to 2.8 × 10^−8^ mm^3^N^−1^m^−1^.

SEM images of wear scars in Figure 10 depict the surface of the ceramic ball in the (Cr_1−x_Al_x_)N/Al_2_O_3_ sliding pair. One can see that the wear scar diameter on the ceramic ball is four or five times smaller than on the steel ball. The highest wear is observed for substrate S1, when the coating hardness is comparable to that of the counter body. The lower the Al content in the (Cr_1−x_Al_x_)N coating, the lower the substrate hardness and wear scar diameter on the counter body surface.

As can be seen from SEM images in Figure 11, the wear scars on the (Cr_1−x_Al_x_)N coating surface appear after the abrasive wear. It should be noted that all the substrates S1–S4 have a low wear rate regardless of the coating hardness.

### 3.6. Friction Coefficient of the (Cr_1−x_Al_x_)N Coating

Friction curves of the (Cr_1−x_Al_x_)N coating measured with 100Cr6 and Al_2_O_3_ counter bodies are shown in Figure 12. In the case with the steel counter body, the friction coefficient does not clearly depend on the coating composition. The jumps observed for the friction coefficient for all the substrates can be associated with the transfer of the counter body material to the coating surface. The smeared material increases the friction coefficient. The correlation of the friction and abrasive wear with the coating composition can be determined by tribology tests with the use of the Al_2_O_3_ counter body. After a short running-in phase of ~15 m, the friction coefficient stabilizes at 0.43 for the coating low in Al and having the lowest wear rate (substrate S4). The running-in phase for the coating high in Al and having the highest wear rate (substrate S1) is much longer (~60 m) and the friction coefficient is higher (0.63).

Table 3 presents the literature data on mechanical and tribological properties of magnetron-sputter-deposited CrAlN coating. Among the coatings obtained by different types of magnetron sputtering, our coating synthesized by short-pulse high-power dual magnetron sputtering has a comparable hardness to other coatings, the highest wear resistance, and the lowest friction coefficient.

Arc-evaporated AlCrN coatings [56] in ball-on-disc experiments against Al_2_O_3_ demonstrated lower wear rate (about 2 × 10^−7^ mm^3^N^−1^m^−1^), but a comparable coefficient of friction (0.63).

### 3.7. Adhesive Strength Measurement

The VDI 3198 Rockwell-C indentation test is often used to evaluate coating adhesion to different substrates [57,58,59]. After indentation, localized plastic deformation occurs near indentation scars, which can cause the coating to fracture. Depending on the fracture type (cracks, laminations, swelling) and size, the damage is classified according to the VDI 3198 standard from HF1 to HF4, where insignificant microcracks and delaminations of the coating near indentation scars are considered as allowable for practical use. HF5 and HF6 indicate unacceptable adhesion with severe delamination around the indentation scar [41]. Rockwell indentation images in Figure 13 describe the respective substrates. One can see microcracks after indentation on substrates S1 and S2 with the hardest coating. Since hard coating is more brittle, plastic deformation induces its cracking. The harder the coating, the longer the microcracks. In general, coatings on substrates S1 and S2 possess good adhesive strength and satisfy HF1 and HF2 strength quality.

Coatings on substrates S3 and S4 possess lower hardness, and no damages are observed near indentation scars. The adhesion of the coating on these substrates is good and satisfies the HF1 strength quality. Thus, the decrease in the Al content in the (Cr_1−x_Al_x_)N coating results in its hardness reduction, but improves the adhesive strength.

## 4. Conclusions

The (Cr_1−x_Al_x_)N coatings with 17 to 54 at.% Al content were deposited onto steel substrates by short-pulse high-power dual magnetron sputtering of Al and Cr targets at a substrate bias voltage. It was demonstrated that the lower Al content in the (Cr_1−x_Al_x_)N coating led to its hardness reduction from 28 to 16 GPa. In tribology testing, the (Cr_1−x_Al_x_)N/100Cr6 sliding pair showed tribochemical wear accompanied by the dominant wear of the counter body, its smearing on the coating surface and oxidation during friction. The (Cr_1−x_Al_x_)N/Al_2_O_3_ sliding pair showed abrasive wear of the surface induced by hard particles of the coating and the counter body. In tribology testing with the ceramic counter body, the (Cr_1−x_Al_x_)N coating possessing lower hardness manifested a lower wear rate due to the less worn counter body and, consequently, a smaller quantity of abrasive particles involved in the friction process. According to the Rockwell-C scale hardness test, the HF1 and HF2 adhesive strength qualities of (Cr_1−x_Al_x_)N coatings with 54 and 45 at.% Al content and high hardness defined their strong interfacial adhesion, whereas the coatings with 31 and 17 at.% Al content and lower hardness showed the adhesive strength quality HF1. In general, (Cr_1−x_Al_x_)N coatings possessed good adhesive strength and satisfied industry standards.

In this work, the (Cr_1−x_Al_x_)N coating was used to demonstrate the advantages of short-pulse high-power dual magnetron sputtering, which can be used to synthesize various hard and wear-resistant coatings based on Ti, Cr, Al and B elements. This will be discussed in our further research.

## Figures and Tables

**Figure 1 materials-15-08237-f001:**
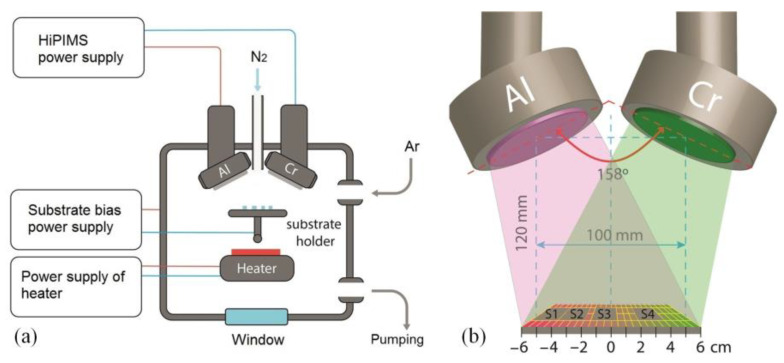
(**a**) Flow-chart of the proposed dual-HiPIMS vacuum system and (**b**) arrangement of S1–S4 substrates on the holder relative to magnetrons.

**Figure 2 materials-15-08237-f002:**
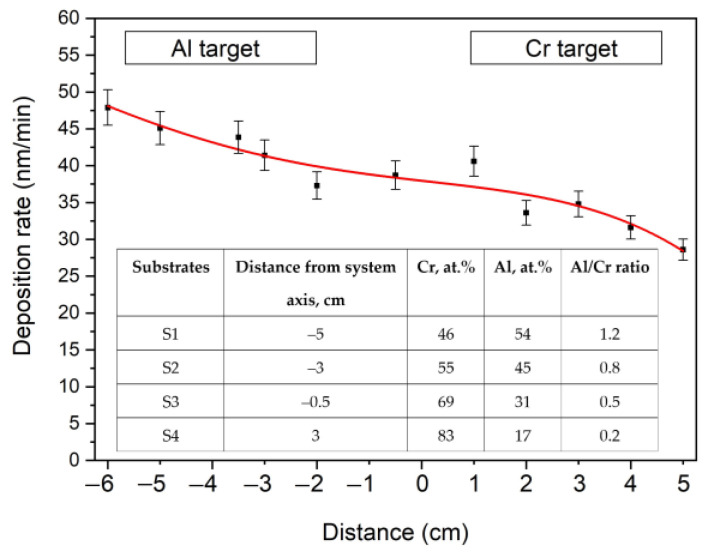
Lengthwise distribution of (Cr_1−x_Al_x_)N coating deposition rate on substrate holder. Table: elemental composition of (Cr_1−x_Al_x_)N coating.

**Figure 3 materials-15-08237-f003:**
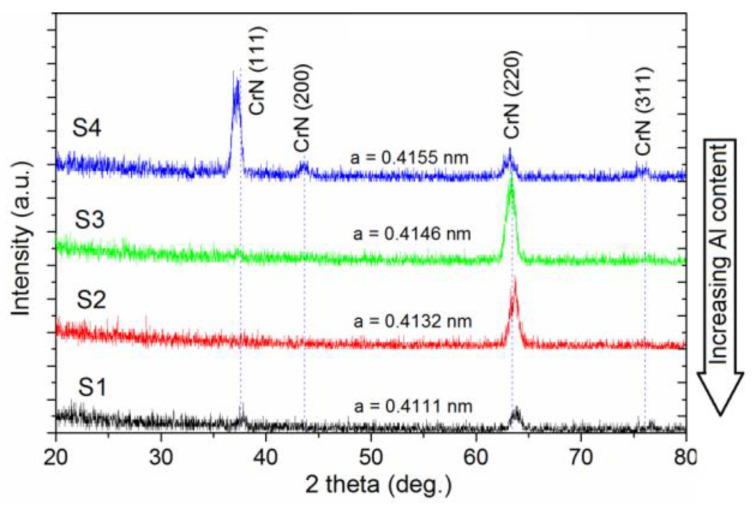
XRD patterns of (Cr_1−x_Al_x_)N coatings.

**Figure 4 materials-15-08237-f004:**
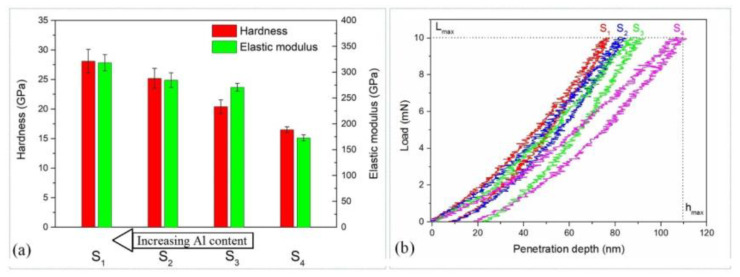
(**a**) Hardness, elastic modulus and (**b**) loading–unloading curves obtained for (Cr_1−x_Al_x_)N coating after nanoindentation measurement.

**Figure 5 materials-15-08237-f005:**
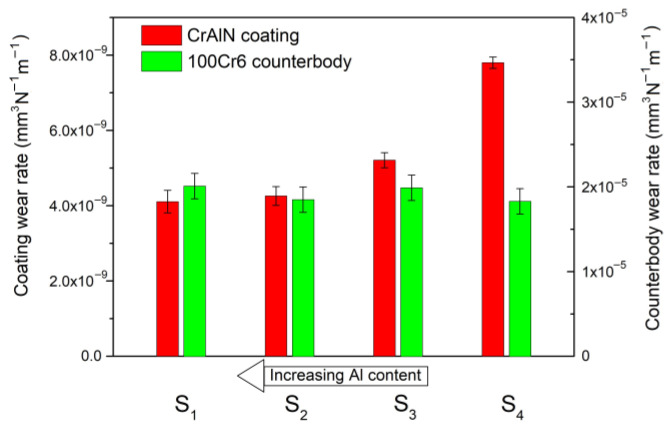
Wear rate of (Cr_1−x_Al_x_)N coating and counter body for (Cr_1−x_Al_x_)N/100Cr6 sliding pair.

**Figure 6 materials-15-08237-f006:**
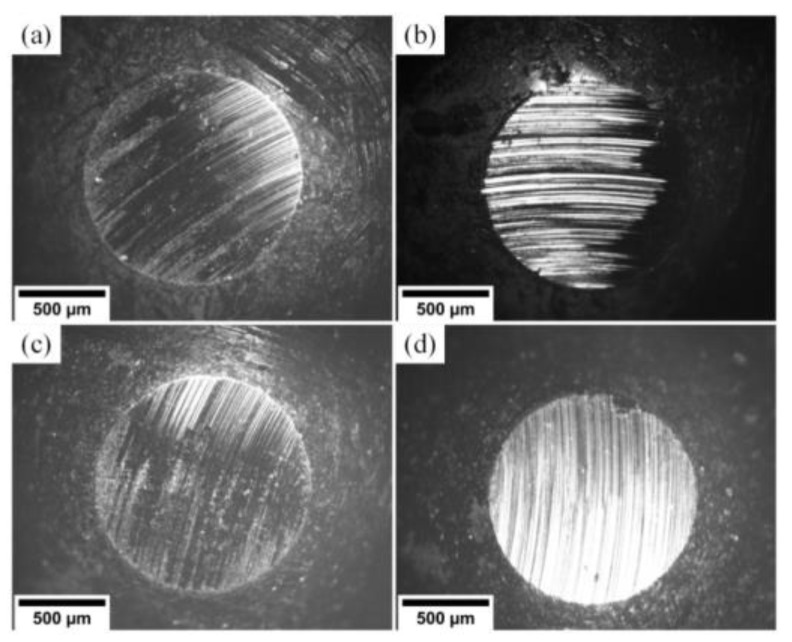
Wear scars on the 100Cr6 ball surface for substrates: (**a**) S1 (Al/Cr = 1.2), (**b**) S2 (Al/Cr = 0.8), (**c**) S3 (Al/Cr = 0.5), (**d**) S4 (Al/Cr = 0.2).

**Figure 7 materials-15-08237-f007:**
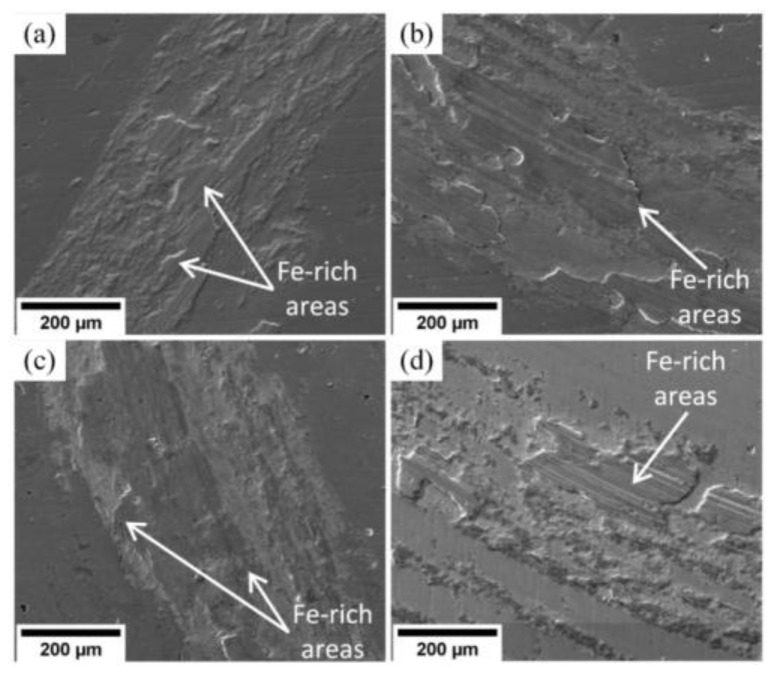
Wear tracks on the surface produced by the 100Cr6 counter body for substrates: (**a**) S1 (Al/Cr = 1.2), (**b**) S2 (Al/Cr = 0.8), (**c**) S3 (Al/Cr = 0.5), (**d**) S4 (Al/Cr = 0.2).

**Figure 8 materials-15-08237-f008:**
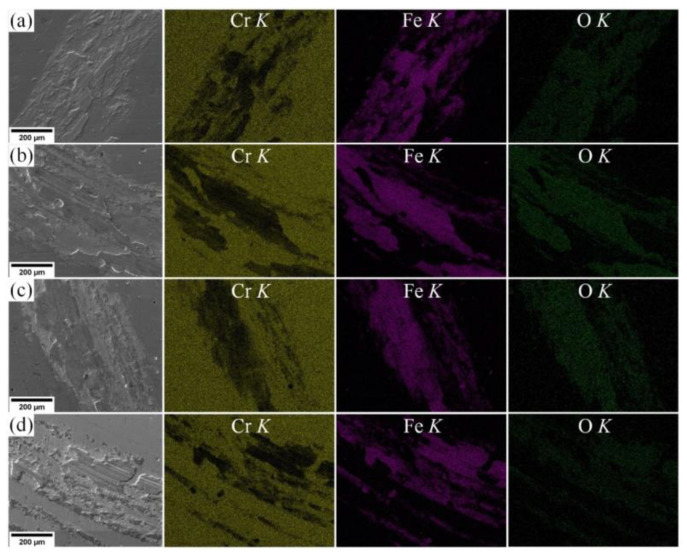
SEM images of wear tracks and distribution of chromium, iron and oxygen in coatings: (**a**) S1 (Al/Cr = 1.2), (**b**) S2 (Al/Cr = 0.8), (**c**) S3 (Al/Cr = 0.5), (**d**) S4 (Al/Cr = 0.2).

**Figure 9 materials-15-08237-f009:**
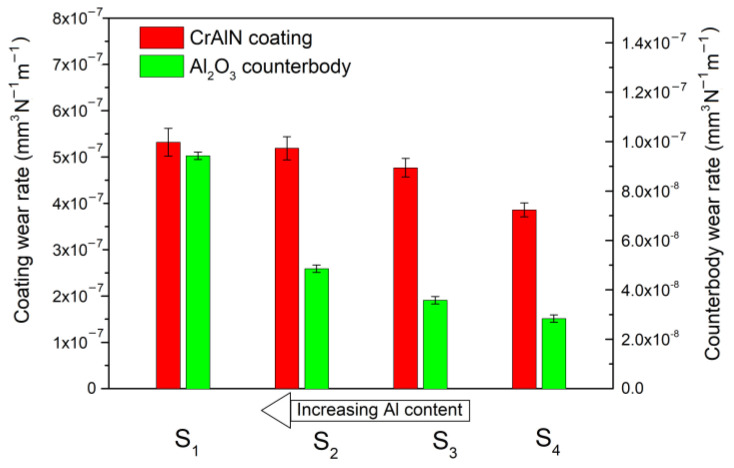
Wear rate of (Cr_1−x_Al_x_)N coating and counter body for (Cr_1−x_Al_x_)N/Al_2_O_3_ sliding pair.

**Figure 10 materials-15-08237-f010:**
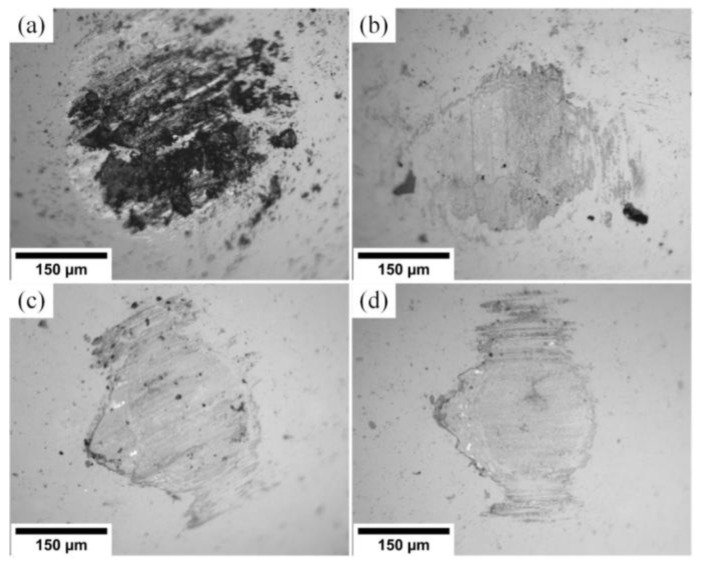
Wear scars on the Al_2_O_3_ ball surface for substrates: (**a**) S1 (Al/Cr = 1.2), (**b**) S2 (Al/Cr = 0.8), (**c**) S3 (Al/Cr = 0.5), (**d**) S4 (Al/Cr = 0.2).

**Figure 11 materials-15-08237-f011:**
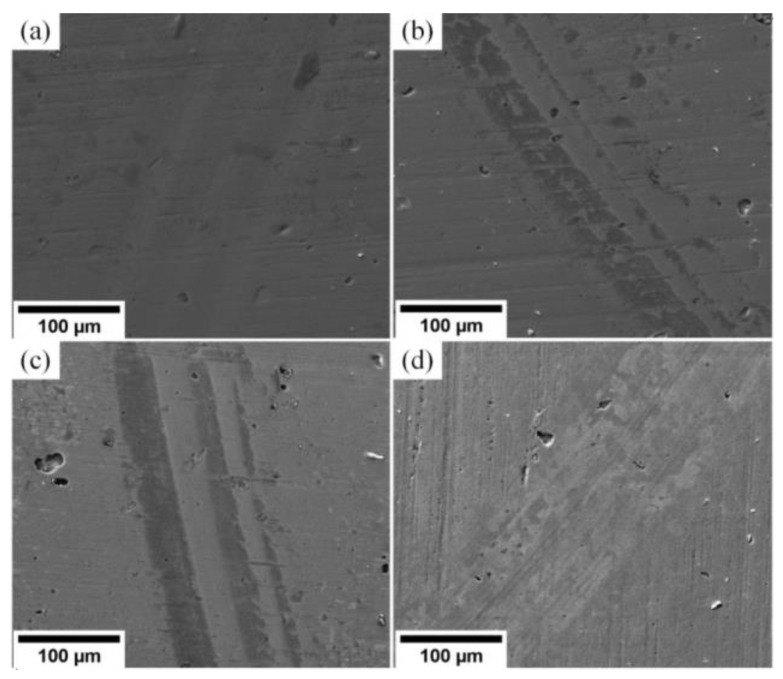
Wear tracks on the surface produced by Al_2_O_3_ counter body for substrates: (**a**) S1 (Al/Cr = 1.2), (**b**) S2 (Al/Cr = 0.8), (**c**) S3 (Al/Cr = 0.5), (**d**) S4 (Al/Cr = 0.2).

**Figure 12 materials-15-08237-f012:**
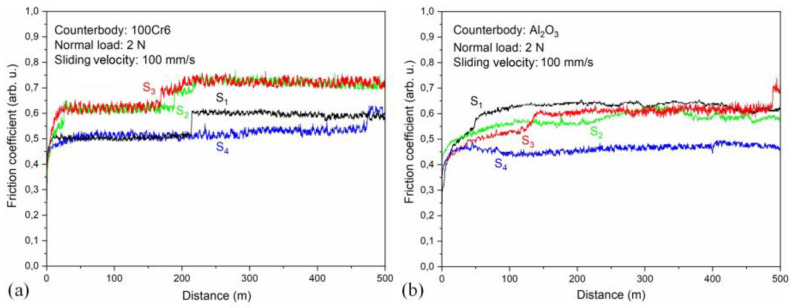
Coefficient of friction as a function of sliding distance measured by ball-on-disc method: (**a**) 100Cr6, (**b**) Al_2_O_3_ counter bodies on the (Cr_1−x_Al_x_)N coating.

**Figure 13 materials-15-08237-f013:**
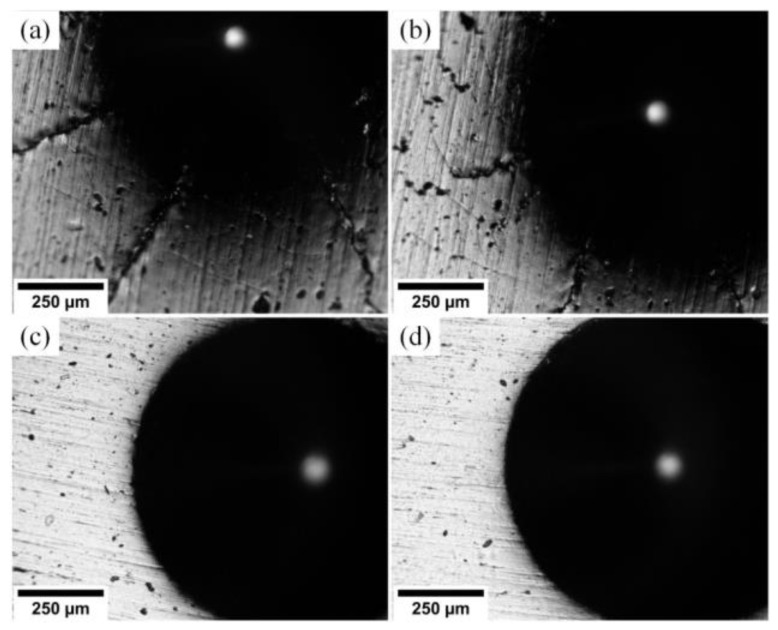
Rockwell indentation images of substrates: (**a**) S1 (Al/Cr = 1.2), (**b**) S2 (Al/Cr = 0.8), (**c**) S3 (Al/Cr = 0.5), (**d**) S4 (Al/Cr = 0.2).

**Table 1 materials-15-08237-t001:** Physical and mechanical properties of counter bodies [40] and substrate.

Counter Body	H, GPa	E, GPa	δ, g∙cm^−3^	υ, a.u.	H^3^/E^2^, MPa
100Cr6	2.7 ± 0.2	185 ± 17	7.8	0.30	0.57
Al_2_O_3_	26 ± 2.3	382 ± 32	3.9	0.24	120.4
Substrate	2.5 ± 0.1	182 ± 7	7.7	-	0.47

*Notation*: H—hardness, E—elastic modulus, δ—density, υ—Poisson’s ratio.

**Table 2 materials-15-08237-t002:** Mechanical parameters of the (Cr_1−x_Al_x_)N coating.

Substrates	H, GPa	E, GPa	H/E	H^3^/E^2^, MPa	W_e_, %
S1	28	318	0.088	219	89.1
S2	25	284	0.089	198	88.5
S3	20	271	0.075	115	74.1
S4	17	173	0.095	150	83.4

**Table 3 materials-15-08237-t003:** Mechanical and tribological properties of magnetron-sputter-deposited CrAlN coating.

Methods	Substrate	Al/Cr Ratio	Hardness (GPa)	Wear Rate (mm^3^N^−1^m^−1^)	COF (a.u.)	Counter Bodies	Literature
RF sputtering	Si, stainless steel	1.4	30.9	-	0.4–0.8	SiC	[18]
DC sputtering	AISI 304	-	-	3.7 × 10^−6^	0.4–0.9	Si_3_N_4_	[22]
DC sputtering	M2 HSS	1	28	-	0.57	Al_2_O_3_	[53]
C-HPMS	Si, cemented carbide	-	35.6	-	0.76	Si_3_N_4_	[24]
HiPIMS/DC co-sputtering	9Cr18 stainless steel	1.3	20.7	8 × 10^−7^	0.5	Al_2_O_3_	[25]
CFUMS	High speed steel	0.2	23	6 × 10^−7^	-	Al_2_O_3_	[15]
CFUMS	AISI 304	0.4	27.4	3.69 × 10^−6^	0.41	WC-Co (6%)	[54]
MF MS *	AISI 420	2	23	4 × 10^−6^	-	Al_2_O_3_	[55]
Short-pulse dual HiPIMS	AISI 430	1.2	28	5.3 × 10^−7^	0.6–0.63	Al_2_O_3_	Our work
		0.2	17	3.9 × 10^−7^	0.42–0.45		

*** MF MS—middle-frequency magnetron sputtering.

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
