# Peer review of "(Cr1−xAlx)N Coating Deposition by Short-Pulse High-Power Dual Magnetron Sputtering"

_materials, 2022, doi:10.3390/ma15228237_

Round 1

Reviewer 1 Report

In the introduction section, the authors claim they propose to deposit CrAlN applying the methodology for the first tiem. After the authors highlight the advantages of the method, they mentioned a second advantage  and a reference from 2008 is included. This is showing that certain aspects of the methodology are relatively old. The same applies for the Fourth advantage, this part included references 35 to 37. This is confusingn and should be clearly explained what is already published and what is new in the present publication.

The coating deposition method needs further and better explanation. The intro is based on the argument that Cr/Ar ratio can be varied during deposition (i.e. graded Al o Cr content), but this is not described in the deposition method. This is also claimed to present an effect on the coating adhesion and should be discussed as stated in the draft.

Basic information on surface and cros-sections of the coatings would be helpful to understand the correlation of the deposition process (that provides less defects) with the microstructure and the properties discussed in the draft.

Figure 2 shows more than 4 samples - the authors should mention which ones are S2 to S4 for clarity.

In the XRD section, the authors mentioned that higher Al-content leads to the 220 orientation, but, Fig. 3 states de opposite. S1 which contains the highest Al-content show only a small peak in the 220 direction, while the S2 and S3 display a different tendency. This sections need clarity in this sense.

Some references are not needed, there are well-known requirements for nanoindentation of coatings. For instance the 10% depth relative to the coating thickness. If you mention this aspect would be enough. Less is more.

Fig 4b is not well-presented since S4 convers the data related to the other ones. Adds not value. If you maintain it, a better way to show the final penetration depth is needed.

The tribology section needs attention when comparing the wear-processes. The AlCrN coating should be compared to similar coatings and wear-conditions, not against TiN. Maybe would be useful to compare the system to other deposition techniques such as arc-PVD due to the defects asociated to this method.

Adhesion testing should be the first coating quality factor. The other techniques would be irrelevant on coatings with low adhesion. Would be better to reorganize the adhesion data on the first sections related to the results and discussion.

Reviewer 2 Report

Dear authors

I read the article with great interest. I believe it is worthy of publication, but it also requires supplying lacking information, discussion, and consideration of several issues. I think it would be worthwhile for the authors to read my suggestions and consider making corrections or at least help me understand some of the points raised. Here is a list of my concerns:

1.      The experiment idea

Changing the aluminum content of the (Cr1−xAlx)N phase provides a spatial location of the substrate concerning Al and Cr magnetrons. This is an attractive solution for the preliminary research stage but is not applicable in industrial practice. In their future work, the authors should try to provide another way to control the Al content. I think it is worthwhile to implement this with different discharge power densities on individual magnetrons.

2.      Page 2, lines 52-54: "(…) it is important to control the elemental composition of coatings directly during the deposition process in order to obtain a gradient composition or multilayer coatings"

This is an excellent and essential statement. Based on their results, can the authors determine how to use the gradient of Al content in (Cr1−xAlx)N phase to obtain gradient coatings with excellent anti-wear properties? This can be completed in the Discussion section.

3.      Page 2, 61-64: "Secondly, two unbalanced magnetrons neighboring each other with opposite polarities can form a closed configuration of magnetic field, extending the plasma to the substrate and increasing the current density on it providing a positive effect on the coating properties"

Are the authors sure that the geometry of the experiment ensures this situation? I have doubts about this. It seems that the substrate is located outside the created "magnetic circuit" between the magnetrons.

4.      Materials and Methods - missing information

Were the substrates heated during the entire deposition process?

How were the magnetrons powered? From one power supply or two? If from one, does that mean they worked simultaneously or alternately?

How was the discharge power measured? Were the 0.5 and 1 kW powers intentionally set, or were they the result of the discharge conditions?

What were the HiPIMS voltage and current peaks?

Was the polarization electrically coupled to the HiPIMS current peak?

5.      Results and discussion

Have the authors done a study of the oxygen content of their coatings? From my experience, it is not easy to ensure the purity of the Al - N system phase formation process due to residual oxygen. As a result, phases are formed from the Al - O - N system.

6.      Fig. 2 and Table 2

I encourage the authors to combine Fig.2 and Tab. 2 into one figure. Under the deposition rate curve, can be placed in the appropriate positions on the scale Distance cumulative bars presenting the Al and Cr content

7.      Other figures

I ask the authors to consider adding arrows to the figures showing the increase in Al content in samples S1...S4. This will make them easier to read

Round 2

Reviewer 1 Report

The distances of the table (insert in Fig. 2) do not exactly match the positions (distances) in Fig. 2. 

Author Response

The figure 2 has been modified. 
